# Sensor Head Temperature Distribution Reconstruction of High-Precision Gravitational Reference Sensors with Machine Learning

**DOI:** 10.3390/s24082529

**Published:** 2024-04-15

**Authors:** Zongchao Duan, Feilong Ren, Li-E Qiang, Keqi Qi, Haoyue Zhang

**Affiliations:** 1School of Fundamental Physics and Mathematical Sciences, Hangzhou Institute for Advanced Study, University of Chinese Academy of Sciences, Hangzhou 310024, China; duanzongchao21@mails.ucas.ac.cn; 2National Space Science Center, Chinese Academy of Sciences, Beijing 100190, China; 3Taiji Laboratory for Gravitational Wave Universe (Beijing/Hangzhou), University of Chinese Academy of Sciences, Beijing 100049, China; 4Xi’an Aerospace Remote Sensing Data Technology Corporation, Xi’an 710054, China; 5Institute of Mechanics, Chinese Academy of Sciences, Beijing 100190, China; 6Research Center of Satellite Technology, Harbin Institute of Technology, Harbin 150001, China

**Keywords:** gravitational reference sensors, temperature reconstruction, simulation, interpolation, machine learning

## Abstract

Temperature fluctuations affect the performance of high-precision gravitational reference sensors. Due to the limited space and the complex interrelations among sensors, it is not feasible to directly measure the temperatures of sensor heads using temperature sensors. Hence, a high-accuracy interpolation method is essential for reconstructing the surface temperature of sensor heads. In this study, we utilized XGBoost-LSTM for sensor head temperature reconstruction, and we analyzed the performance of this method under two simulation scenarios: ground-based and on-orbit. The findings demonstrate that our method achieves a precision that is two orders of magnitude higher than that of conventional interpolation methods and one order of magnitude higher than that of a BP neural network. Additionally, it exhibits remarkable stability and robustness. The reconstruction accuracy of this method meets the requirements for the key payload temperature control precision specified by the Taiji Program, providing data support for subsequent tasks in thermal noise modeling and subtraction.

## 1. Introduction

The detection of gravitational waves can offer new insights into the universe’s origins, formations, and evolution. Several space missions have been proposed to detect gravitational waves, including the Laser Interferometer Space Antenna (LISA) mission [1], the Tianqin mission [2], and the Taiji mission [3]. A space-borne gravitational wave detector, consisting of three identical spacecraft forming a near-equilateral triangle that exchanges laser beams over millions of kilometers via long arms, aims to measure gravitational waves over a broad band at low frequencies, from about 0.1 mHz to 1 Hz [4,5,6]. By using precision laser interferometry, gravitational waves can be detected by measuring the changes in distance between test masses (TMs) millions of kilometers apart. These TMs are protected from non-gravitational influences by the spacecraft and move along the geodesic, providing an ideal inertial reference. In addition to gravitational wave detection missions, gravitational reference sensors and accelerometers are widely used in space microgravity experiment missions, satellite gravity recovery missions, and other space missions [7,8].

The gravitational reference sensor (GRS) includes the TM, enclosed in an electrode housing (EH), the launch lock mechanism, and the charge control system. The electrode housing, with the electrodes inside the housing, is used to read out the position of the test mass relative to the housing [9]. TMs are susceptible to disturbances caused by fluctuations in magnetic fields, electric fields, temperature, and the self-gravity effect [10,11,12,13]. Among these disturbances, temperature distribution and thermal stability have a noteworthy impact on the performance of the GRS or accelerometer. For instance, the temperature gradient affects the accelerometer of the GRACE-FO satellite [14]. Variations in temperature gradients can lead to fluctuations in accelerometer linearity and angular measurements, thereby impacting the recovery results of Earth’s gravitational field. In the µSCOPE mission [15], researchers investigated the impact of temperature gradients on the detection of equivalence principle (EP) violations. Even if µSCOPE was constructed according to stringent requirements, including material and density selection for the TM, the design of the temperature control systems, and the precise calibration of sensors and measurement devices, temperature gradients from the low-Earth orbit of µSCOPE pose challenges. These temperature gradients amplify the radiometer effect, making it challenging to distinguish this from potential EP violation signals. LISA Pathfinder (LPF) has demonstrated that temperature fluctuations can severely affect the sensitivity of instruments, including the GRS and optical metrology subsystem (OMS) [16]. LPF devised a model linking thermal disturbances to interferometer readings by establishing a transfer function connecting the heat source to the temperature at a specific point, using a vector-fitting method to determine this thermal transfer function [17,18]. Researchers from the Taiji mission team observed that temperature noise can overwhelm the measurement signal in the low-frequency band, which would limit the detection sensitivity of GRS [19]. Measurement data from the Taiji-1 mission show that there is a strong correlation between temperature sensor readings and GRS channel drive voltages [20]. In order to mitigate the effects of temperature gradients on gyroscopes, G-Probe B operates at low pressures and temperatures, with the gyro housed in an environment composed of high-thermal conductivity materials to minimize the temperature gradient difference across its surface [21]. In summary, temperature gradients and temperature fluctuations in the EH wall cause thermal acceleration noise acting on the TM [22], which negatively impacts the performance of GRS. Three distinct thermal effects have been identified [23]: thermal radiation pressure, the thermal radiometer effect, and asymmetric outgassing.

Temperature sensors are placed around the inertial sensors to monitor temperature changes at the measurement points. Common types of temperature sensors include thermocouples, thermistors (such as NTC and PTC thermistors), semiconductor junctions (such as PN junctions and Schottky diodes), optical fibers, and capacitive sensors [24]. Each of these temperature sensors has its specific temperature measurement range and accuracy. High-end temperature sensors, such as Schottky Diode sensors based on silicon carbide (SiC), are capable of operating in high-temperature environments. These sensors exhibit superior temperature stability and lower power consumption [25]. By incorporating a junction termination extension (JTE) layer [26], their high voltage resistance can be enhanced. Such sensors are suitable for applications requiring precise temperature control and high-stability demands. Real-time measurements from the temperature sensors can be used to reconstruct the temperature distribution across the EH and evaluate the thermal noise of the inertial sensors. Due to the limitation of the number and location of the temperature sensors [27,28,29], an accurate interpolation algorithm is needed [30]. With the development of artificial intelligence technology, neural network algorithms are now applied in gravitational wave detection missions, such as signal detection and parameter estimation [31,32]. Traditional interpolation algorithms perform poorly when the size of the interpolation region is excessive [33]. The LPF team proved the effectiveness of utilizing BP neural networks to reconstruct the magnetic field of sensor heads [34].

We propose an algorithm utilizing XGBoost-LSTM (XL) to reconstruct the EH surface temperature distribution from a limited number of temperature sensors. Traditional interpolation methods often depend heavily on the spatial correlation between temperature sensors and the desired interpolation positions [35]. The temperature values at the boundary positions of the EH are obtained through linear interpolation between temperature sensor readouts. The use of linear terms with spatial co-ordinates cannot fully capture the temperature gradient details on the surface of the EH. In contrast, our method can efficiently learn the latent associative information between temperature sensors and the areas to be predicted. We find that our algorithm has advantages over unimproved neural networks, including the BP neural network and the long short-term memory (LSTM) network. These networks lack sufficient physical information to describe the entire temperature field. Specifically, when predicting a specific area’s temperature, performance may falter elsewhere—especially if the BP neural network’s output dimensions surpass its input dimensions. For a detailed surface temperature distribution of the EH, providing varied temperature sensor weight combinations for different areas is essential. Therefore, we designed the XGBoost-LSTM algorithm to solve this problem. The proposed algorithm makes the following three contributions to the reconstruction of the surface temperature of the sensor head:The algorithm’s dynamism allows it to reconstruct temperature at any point on the EH surface using temperature sensor data with variable weights.Compared to the BP neural network, the algorithm is less dependent on the number of temperature sensors.The intermediate output of the algorithm presents the weight information of the temperature sensors, which can be determined through further experiments to ascertain the optimal number and placement of temperature sensors.

The primary content of this study is depicted in Figure 1.

The remainder of this article is organized as follows. In Section 2, we detail the impact of temperature noise generated by various thermal effects on the performance of GRS. In Section 3, we introduce two simulation schemes, and the subsequent reconstruction algorithms are based on the simulation data from this part. In Section 4, we introduce the implementation principle of XGBoost-LSTM. We also briefly introduce existing algorithms, which were utilized for comparative experiments. Finally, the metrics used in the evaluation of these algorithms during the experiments are introduced. In Section 5, we present the results of experiments based on ground simulation data and on-orbit simulation data. In Section 6, we summarize the work we have completed and the work that is currently underway.

## 2. Analysis of Temperature Noise Impact

The space-borne gravitational wave detection mission requires high GRS accuracy. The TM is subjected to physical environmental factors such as a magnetic field, electric field, and temperature gradient, which result in acceleration noise and are read out by the interferometer, interfering with scientific measurements. Regarding the temperature effect, the TM is not significantly affected by external temperature fluctuations [36]. Therefore, temperature gradients primarily affect the EH, causing forces and torques on the TM.

In order to ensure the achievement of low-frequency space-borne gravitational wave detection scientific objectives in the Taiji project, strict requirements must be imposed on the residual acceleration noise of the TM. The residual acceleration total noise budget allowed by the Taiji program is as follows [37]: (1)$$S_{\delta a,Taiji}^{1/2}\left(\omega \right)⩽3\times 10^{-15}\left[1+{\left(\frac{\omega /2\pi }{3\;\mathrm{mHz}}\right)}^{2}\right]{\mathrm{ms}}^{-2}{\mathrm{Hz}}^{-1/2}.$$

Three thermal effects have been identified that adversely affect the TM: the thermal radiation pressure, the thermal radiometer effect, and asymmetric outgassing. These effects originate from the temperature gradient across the EH surface, leading to relative displacement and subsequently influencing the TM.

### 2.1. Thermal Radiometer Effect

When the surfaces of the TM are at different environmental temperatures, the forces exerted by the residual gas on the surfaces of the TM differ, which is called the radiometer effect. The acceleration noise caused by the radiometer effect can be calculated as follows: (2)$${\tilde{a}}_{rd}=\frac{PA}{4T_{0}m_{t}}{\tilde{\delta }}_{T}.$$
where ${\tilde{a}}_{rd}$ represents the acceleration noise caused by the radiometer effect, *P* and ${\tilde{\delta }}_{T}$ refer to the differences in pressure and temperature between the sides of the TM, *A* represents the surface area of the TM, $T_{0}$ represents the temperature of the EH, and $m_{t}$ represents the mass of the TM.

### 2.2. Thermal Radiation Pressure

Owing to the temperature and temperature gradients, thermal radiation is generated within the sensor head. This thermal radiation imposes a radiation pressure on the TM surface, resulting in noise on the TM. The aforementioned issues can be seen as problems based on the transfer of thermal photon momentum and energy. Accordingly, the magnitude of the noise caused by thermal radiation pressure can be deduced: (3)$${\tilde{a}}_{tr}=\frac{8\sigma A}{3cm_{t}}T_{0}^{3}{\tilde{\delta }}_{T}.$$
where ${\tilde{a}}_{tr}$ represents the acceleration noise caused by thermal radiation pressure, σ is the Stefan–Boltzmann constant, and *c* represents the speed of light. Figure 2 illustrates the effects of the thermal radiometer and radiation pressure.

By incorporating the design parameters of the GRS (refer to Equation (Equation 30)) and presuming a maximum temperature difference of $10^{-5}$ K in the sensor head, we deduce that the magnitudes of both the thermal radiometer effect and the thermal radiation pressure effect are approximately of the order of $10^{-16}$
${\mathrm{ms}}^{-2}$.

### 2.3. Asymmetric Outgassing

Due to temperature differences and thermal expansion, uneven gas flow exists in different parts. This phenomenon leads to gas molecules detaching from the surface in a low-pressure environment. The gas outflow from the surface can be simulated using the temperature activation law: (4)$$Q\left(T\right)=Q_{0}e^{\left(-\Theta /T\right)},$$
where $Q_{0}$ is a pre-exponential factor, and Θ represents the required activation temperature for the gas under consideration. This simplified model considers only one type of gas molecule released from the inner surface of the EH, and there is no adhesion when the molecules released from the inner surface of the EH collide with the TM. The presence of a temperature gradient $\Delta T_{x}$ leads to different rates of gas release.
(5)$$\Delta Q\left(T\right)\approx Q\left(T_{0}\right)\left(\Theta /T_{0}\right)\left(\Delta T_{x}/T_{0}\right),$$

The results have led to different pressures in various aspects of test quality: (6)$${\tilde{a}}_{og}\approx \frac{1}{m_{t}}A\frac{\Delta Q\left(T_{0}\right)}{C_{eff}}\frac{\Theta }{T_{0}}\frac{\Delta T_{x}}{T_{0}}.$$

The parameter $C_{eff}$ is determined by the geometric shape of the sensor head and is related to the distribution of gaps between the sensor head and the holes in the inner wall.

Research has indicated that the estimation of parameters associated with asymmetric outgassing carries uncertainty, yet its magnitude is similar to the other two effects [38,39]. Therefore, this effect is ignored in the subsequent total noise calculation. Because the thermal radiometer effect and the thermal radiation pressure have the same noise source, $\tilde{\delta_{T}}$, their acceleration noises are fully correlated and can be accumulated. The following formula is used to estimate the temperature control accuracy of the key payload: (7)$$S_{\tilde{a}}^{1/2}\left(\omega \right)=\left[\frac{PA}{4T_{0}m_{t}}+\frac{8\sigma A}{3cm_{t}}T_{0}^{3}\right]S_{{\tilde{\delta }}_{T}}^{1/2}\left(\omega \right).$$

According to the budget given by Equation (Equation 1), the Taiji project sets the target of the temperature control accuracy of the key payload at 100 μ${\mathrm{KHz}}^{-1/2}$. In future experiments involving the reconstruction of the temperature field on sensor heads, we aim to achieve a temperature prediction accuracy at this level.

## 3. Simulation

This section primarily introduces two types of simulation experiments: ground-based tests and on-orbit tests.

### 3.1. General Description of Simulation

We established a simplified EH model composed of aluminum and sapphire, where the minor structures, such as screws and chamfers, have been streamlined. This was carried out to maintain the model’s primary geometric features while minimizing computational demand. The length, width, and height of the simplified EH model are 78, 75.8, and 76 mm, respectively (excluding the electrode cover), and the thickness of the electrode cover is 5 mm. In the simulation, the used material properties align with their actual characteristics. The geometric design of the EH is shown in Figure 3.

For the simulations of the ground-based experiment, we designed two sets of heat sources, each consisting of two units, and positioned them on two opposing sides of the EH. We adopted four heaters for periodic heating at a power level of 0.01 watt, with each heating period lasting for 1000 s. Additionally, we placed eight temperature sensors at diagonal positions on all four sides of the EH to collect the simulated hardware temperature data.

For the simulations of the on-orbit experiment, we arranged four heat sources around the EH, each at a distance of 6–8 cm, to emulate the interference from multiple heat sources in a real environment. Each of these four heaters operated based on its own unique random seed, with heating durations ranging from several tens to several thousands of seconds and the heating power fluctuating between a few hundredths of a watt to several tenths of a watt. The heat was transmitted to the EH through thermal radiation. We positioned four temperature sensors at the base of the EH, each at a distance of 2–3 cm, encircling its perimeter. This positioning is depicted in Figure 4.

We partition the cubic surface of the simplified EH and used a discrete approach to represent the temperature gradient relationship on the surface of the EH. The division is shown in Figure 5.

#### 3.1.1. Ground-Based Test Simulation

In the ground-based test simulation design, we conducted a study on the thermal conduction characteristics of the EH using solid thermal modules. The simulation process mainly involves solving the solid heat–conduction equation to calculate the distribution of the surface temperature of the EH. This equation can be mathematically expressed as follows:(8)$$\nabla \cdot \left(\lambda \nabla T\right)+Q=\rho C\frac{\partial T}{\partial t}.$$
where λ represents the thermal conductivity of the EH, *T* represents the temperature distribution, *Q* is the heat-source term, ρ represents the density of the EH, and *C* represents the heat capacity. This equation expresses the energy-conservation law in the process of heat conduction.

Owing to the design characteristics of the EH, such as non-uniformity, anisotropy, and nonlinearity, the heat transfer equation was appropriately adjusted and modified during the solution process according to the grid division.

#### 3.1.2. On-Orbit Test Simulation

In reality, multiple heat sources of differing powers may exist, which could be electronic components or other factors. By randomly initializing the heat source to heat the EH and employing the radiation heat transfer and thermal conduction equations, a heat radiation conduction model simulating on-orbit situations can be established. According to the Stefan–Boltzmann law, the radiant power of heat is directly proportional to the fourth power of an object’s surface temperature: (9)$$Q_{rad}=\sigma \cdot A\cdot \left(T_{E}^{4}-T_{0}^{4}\right),$$

In this equation, $Q_{rad}$ represents the surface radiative power, σ is the Stefan–Boltzmann constant, “*A*” represents the surface area, $T_{E}$ represents the surface temperature of the EH, and $T_{0}$ represents the environment temperature.

Additionally, according to Fourier’s law, the heat flux density is directly proportional to the temperature gradient.
(10)q=−λ∇T,

In this context, *q* represents the heat flux density, λ represents the thermal conductivity, and ∇T represents the temperature gradient.

When conducting thermal radiation heat transfer calculations, it is necessary to consider the mutual influence between heat conduction and thermal radiation. The coupled equation can be obtained by coupling the heat conduction equation and the thermal radiation heat transfer equation: (11)$$\rho C\frac{\partial T}{\partial t}-\nabla \cdot \left(\lambda \nabla T\right)=\sigma \cdot A\cdot \left(T_{E}^{4}-T_{0}^{4}\right).$$
where ρ represents the density of the EH, *C* represents the specific heat capacity, and *t* represents time.

These equations collectively describe the processes of heat conduction and surface-to-surface radiation heat transfer, coupling these two effects through the evolution of the temperature field. This allowed us to simulate the on-orbit conditions of the sensor head.

### 3.2. Simulation Results

This section presents the results of two types of simulation experiments.

#### 3.2.1. Ground-Based Test Simulation Data

We used symmetric heating plates to conduct solid heat transfer on the surface of the EH, and the heating process of the heat source power function and each temperature sensor are shown in Figure 6. We set the surface temperature fluctuation of the EH from 293.150 to 293.216 K in order to evaluate the reconstruction accuracy of the algorithm under such a large temperature difference.

As shown in Figure 7, we can see this process and visually observe the temperature distribution of the EH surface changing with the alternating heat source.

#### 3.2.2. On-Orbit Test Simulation Data

During on-orbit testing, we applied thermal radiation to the EH by simulating realistic complex heat sources. In contrast to the ground-based testing, temperature sensors were placed around the sensor head, with the farthest temperature sensor, O_T3, positioned 3 cm from the outer side of the EH. We set up four groups of heat sources to heat the EH, with random processing applied to each group. As shown in Figure 8, the surface temperature of the EH exhibited a gradient change, and the surface temperature fluctuated between 293.150 and 293.177 K.

In the on-orbit simulation experiment, four heat sources were used to heat the EH. Figure 9 shows the simulation process.

Typically, the temperature fluctuations of the key payload in orbit are not so large, and the positions of the thermometers are not all concentrated at the bottom of the EH. In the on-orbit simulation test, relatively extreme situations were investigated. Considering the most unfavorable situations, we simulated larger temperature differences and more separation temperature sensors, which increased the difficulty of reconstruction. Although these factors reduced the reconstruction accuracy, the result of on-orbit reconstruction meets our task requirements. These will be demonstrated in the Results section.

## 4. Methods

This section introduces the design principles of XGBoost-LSTM, starting with an introduction to the two basic modules, XGBoost and LSTM, followed by an introduction to the principles of model fusion. Finally, a simple description of the evaluation indicators and related comparison methods is also provided.

### 4.1. XGBoost-LSTM Algorithm

This method utilizes XGBoost to perform feature importance extraction on the original temperature sensor data (the first training process); then, feature crossing is performed by multiplying the extracted weights by the original data element-by-element, yielding scaled features rich in cross information. The scaled features were input into LSTM for further learning (the second training). This not only enhanced the weight sensitivity of the neural network for predicting various areas of the sensor head but also exploited the sequence memory ability of LSTM to reconstruct the target area temperature from two dimensions: the position of the temperature sensor and the historical readings. Therefore, XGBoost-LSTM is not two independent prediction models, nor is it a simple combination of two models. Instead, through the ingenious cross-processing of the intermediate data, self-adaptive weight adjustment is achieved in the learning process via collaboration between the models. The temperature reconstruction process of the algorithm is shown in Figure 10.

#### 4.1.1. XGBoost

Extreme gradient boosting (XGBoost) is an ensemble learning algorithm based on boosting [40], which combines multiple weak classifiers to form a strong classifier. During the iteration process, each newly generated tree fits the residual of the previous tree. A larger number of iterations corresponds to a higher training accuracy. Generally, decision trees are used as weak classifiers. A decision tree is composed of nodes, with each node representing a feature and a specific feature value. The algorithm constructs a decision tree based on the feature values in the training data and uses the boosting technique to integrate multiple decision trees. The objective function consists of error terms and regularization terms: (12)$$obj=\sum_{i=1}^{n}l\left(y_{i},\hat{y_{i}}\right)+\sum_{k=1}^{K}\Omega \left(f_{k}\right),$$
where obj denotes the final objective to be minimized. The first term in the equation represents the accumulated loss over all instances, with *n* being the total number of instances. For each instance, *i*, *l* denotes the loss function, which is defined as the difference between the actual label, $y_{i}$, and the predicted label, $\hat{y_{i}}$. The second term is the regularization term used to prevent overfitting, where *K* represents the total number of trees used in the model. For each tree *k*, $\Omega \left(f_{k}\right)$ gives the complexity of the tree, $f_{k}$. For the *t*-th learning tree, each generation requires the continued fitting of the residual from the previous iteration: (13)$${\hat{y}}_{i}^{\left(t\right)}={\hat{y}}_{i}^{\left(t-1\right)}+f_{t}\left(x_{i}\right),$$

This equation updates the prediction for the *i*-th instance at each step or iteration (*t*) by adding the prediction of the new base learner ($f_{t}$). In other words, the new prediction is the old prediction plus the prediction made by the new model (tree). This is reflective of the additive and iterative nature of boosting algorithms such as XGBoost. The objective function is rewritten such that each learning tree tends to be optimal: (14)$$L^{\left(t\right)}=\sum_{i=1}^{n}l\left(y_{i},{\hat{y}}_{i}^{\left(t-1\right)}+f_{t}\left(x_{i}\right)\right)+\Omega \left(f_{t}\right),$$

In order to find the $f_{t}$ that minimizes the objective function, Taylor expansion was used to expand it around $f_{t}=0$: (15)$$L^{\left(t\right)}≃\sum_{i=1}^{n}\left[l\left(y_{i},{\hat{y}}_{i}^{\left(t-1\right)}\right)+g_{i}f_{t}\left(x_{i}\right)+\frac{1}{2}h_{i}f_{t}^{2}\left(x_{i}\right)\right]+\Omega \left(f_{t}\right),$$

Here, $L^{\left(t\right)}$ is the new objective function at the *t*-th iteration of the boosting process, $g_{i}$ represents the first derivative, and $h_{i}$ represents the second derivative.
(16)$$\left\{\begin{array}{c} g_{i}=\partial_{{\hat{y}}^{\left(t-1\right)}}l\left(y_{i},{\hat{y}}_{i}^{\left(t-1\right)}\right)\\ h_{i}=\partial_{{\hat{y}}^{\left(t-1\right)}}^{2}l\left(y_{i},{\hat{y}}_{i}^{\left(t-1\right)}\right), \end{array}\right.$$

By discarding the error values of the previous t−1 trees, the loss values of each sample were accumulated. At this point, the objective function is
(17)$$L^{\left(t\right)}=\sum_{i=1}^{n}\left[g_{i}f_{t}\left(x_{i}\right)+\frac{1}{2}h_{i}f_{t}^{2}\left(x_{i}\right)\right]+\Omega \left(f_{t}\right),$$

The process of accumulating all the samples in the same leaf node is expressed as follows:(18)$$\begin{array}{cc} obj^{\left(t\right)} & ≃\sum_{i=1}^{n}\left[g_{i}f_{t}\left(x_{i}\right)+\frac{1}{2}h_{i}f_{t}^{2}\left(x_{i}\right)\right]+\Omega \left(f_{t}\right)\\ \; & =\sum_{i=1}^{n}\left[g_{i}w_{q}\left(x_{i}\right)+\frac{1}{2}h_{i}w_{q}^{2}\left(x_{i}\right)\right]+\gamma T+\lambda \frac{1}{2}\sum_{j=1}^{T}w_{j}^{2}\\ \; & =\sum_{j=1}^{T}\left[\left(\sum_{i\in I_{j}}g_{i}\right)w_{j}+\frac{1}{2}\left(\sum_{i\in I_{j}}h_{i}+\lambda \right)w_{j}^{2}\right]+\gamma T. \end{array}$$
where $w_{q}\left(x_{i}\right)$ is the weight of the *q*-th leaf node that the *i*-th instance falls into, γT limits the number of leaf nodes, *T*, in a tree to control its complexity, and $\lambda \frac{1}{2}\sum_{j=1}^{T}w_{j}^{2}$ reduces the leaf weights to avoid overfitting, where λ is the L2 regularization term, and $w_{j}$ is the weight of the *j*-th leaf node. Thus, the objective function becomes a quadratic function of the leaf-node weights, which can be solved using a computer.

We pre-trained temperature sensors using XGBoost with the goal of obtaining the weights of the sensors corresponding to different areas. In XGBoost, two metrics—gain and cover—are commonly used to calculate feature importance. These metrics are employed to quantify how much each feature can reduce the objective function when used as a classifier; thus, the importance of the features is estimated. Gain represents the average increase in a prediction made by a feature across all trees. This parameter reflects the splitting ability of the feature at each node, and a stronger splitting ability at each node contributes more to the final prediction. Cover represents the average coverage of samples by a feature across all trees. This parameter reflects the coverage ability of the feature for the model; a feature that influences more samples contributes more to the final prediction. The calculation formulas for gain and cover are as follows: (19)$$\;\mathrm{Gain}\;=\frac{1}{2}\left[\frac{G_{L}^{2}}{H_{L}+\lambda }+\frac{G_{R}^{2}}{H_{R}+\lambda }-\frac{{\left(G_{L}+G_{R}\right)}^{2}}{H_{L}+H_{R}+\lambda }\right]-\gamma .$$
(20)$$\;\mathrm{Cover}\;=\frac{1}{2}\left[\frac{GH_{L}}{H_{L}+\lambda }+\frac{GH_{R}}{H_{R}+\lambda }\right].$$

Here, $G_{L}$ and $G_{R}$ represent the sums of the first-order derivatives of the left and right subtrees, respectively, and $H_{L}$ and $H_{R}$ represent the sums of the second-order derivatives of the left and right subtrees, respectively. λ is the regularization parameter used to control the complexity of the model. γ is another regularization parameter known as the minimum split loss, which controls the required profit value for the minimum split. $GH_{L}$ and $GH_{R}$ represent the sum of the gradients of the left and right subtrees and the product of the sums of the second-order derivatives of the left and right subtrees, respectively.

#### 4.1.2. LSTM

Long short-term memory (LSTM) [41] is an improved version of the recurrent neural network (RNN). It adds a memory cell state to each memory neuron in its network to reduce the rate of information loss compared with traditional RNNs, thereby significantly easing the problem of gradient vanishing [42]. Additionally, it employs three gate structures—the forget gate, input gate, and output gate—to selectively remember the modified parameters of the error function during gradient descent, thereby achieving outstanding sequence memory capabilities. Figure 11 shows a cell unit diagram of LSTM.

The function of the forget gate is to concatenate the input, $x_{t}$, of the current timestep and the hidden state, $h_{\left(t-1\right)}$, of the previous timestep and then process the result through a fully connected layer before applying the sigmoid activation function. The output values of the sigmoid function are between 0 and 1, and they determine how much of the past information needs to be forgotten. When the output of the forget gate is close to 0, most of the past information will be forgotten. Conversely, if the output of the forget gate is close to 1, most of the past information will be retained. The forget gate value is applied to the previous layer’s cell state, representing the amount of past information that is forgotten.
(21)$$f_{t}=\sigma \left(W_{f}\cdot \left[h_{t-1},x_{t}\right]+b_{f}\right).$$

In this equation, $f_{t}$ represents the forget gate at the current timestep, $W_{f}$ is the weight matrix of the forget gate, $x_{t}$ represents the input at the current timestep, $h_{\left\{t-1\right\}}$ represents the output at the previous timestep, $b_{f}$ is the bias term of the forget gate, and σ represents the sigmoid function.

The input gate is another crucial component of LSTM. This step determines which new information should be remembered and added to the cell state, and it is divided into two sub-steps. The first sub-step employs a neural network layer using the sigmoid activation function to determine how much data should be updated. The second sub-step employs another neural network layer using the tanh activation function to generate a candidate vector. This candidate vector is then multiplied by the output vector from the sigmoid function to determine which updated information should be added to the cell state. The dot-product operation is performed element-wise, resulting in an updated cell state, which is the cell state from the previous timestep plus new information extracted from the current timestep input.
(22)$$i_{t}=\sigma \left(W_{i}\cdot \left[h_{t-1},x_{t}\right]+b_{i}\right),$$
(23)$${\tilde{C}}_{t}=\tanh\left(W_{C}\cdot \left[h_{t-1},x_{t}\right]+b_{C}\right),$$
(24)$$C_{t}=f_{t}⊙C_{t-1}+i_{t}⊙{\tilde{C}}_{t}.$$

Here, $i_{t}$ represents the input gate at the current timestep, $W_{i}$ is the weight matrix of the input gate, $b_{i}$ is the bias term of the input gate, and *C* represents the cell.

The output gate is used to determine the output of LSTM by combining the current cell state, the input of the current layer, and the hidden state of the previous layer. Similar to the forget gate and the input gate, it consists of a fully connected layer and two activation functions. Both the input of the current layer and the hidden state of the previous layer are passed through the fully connected layer and then combined with the current cell state through a neural network with the tanh activation function to obtain a vector. This vector is then multiplied by another vector generated by the sigmoid function to generate the final output. The output gate determines which information should be output at the current timestep, i.e., the hidden state of the previous layer plus the new information contained in the cell state at the current timestep.
(25)$$o_{t}=\sigma \left(W_{o}\cdot \left[h_{t-1},x_{t}\right]+b_{0}\right),$$
(26)$$h_{t}=o_{t}⊙\tanh\left(C_{t}\right).$$

Here, $o_{t}$ represents the output gate at the current timestep, $W_{o}$ is the weight matrix of the output gate, and $b_{0}$ is the bias term of the output gate.

#### 4.1.3. Algorithm Implementation Principles

As shown in Figure 12, XGBoost-LSTM (XL) is an adaptive weighted combination method that utilizes element-wise multiplication for weighting. It is neither a simple sum of individual models nor a “secondary prediction” from XGBoost to LSTM. XGBoost and LSTM both have high prediction accuracies for nonlinear and non-stationary data, and their principles differ significantly. Through reasonable improvements, XGBoost is used to fully explore useful information in temperature sensor data, and the artificially assigned weights of temperature sensors at specific prediction areas are further learned via LSTM, significantly increasing the convergence speed and learning efficiency of LSTM.

In summary, by exploiting the characteristics of tree models, the proposed algorithm achieves accurate adaptive temperature-field reconstruction. By selecting temperature sensor data with a stronger impact on target value prediction for the prediction areas, the weight information is uniformly processed and combined with temperature-sensor values to create higher-order cross features. The processed data are then input into the LSTM neural network for further learning; thus, the mapping relationship between temperature sensors and the surface temperature of the sensor head is learned. The algorithm learns to be read by the neural network from feature weight learning, achieving an end-to-end automated training mode.

### 4.2. Other Methods

In the temperature-field reconstruction experiment, we compared the proposed XL algorithm with several baseline models, including the BP neural network and polynomial interpolation (PI). These baseline algorithms are described below.

The primary principle of the BP algorithm involves minimizing the network’s error function by iteratively updating its parameters. This process starts with a forward pass, where input data are propagated through the network, resulting in an output prediction. This is followed by a backward pass, during which the computed error is propagated backward from the output layer to the input layer, allowing for the determination of the gradients associated with each parameter.

The principles of PI are based on the interpolation problem, which involves finding a polynomial function that exactly matches a set of known data points. The fundamental assumption behind this process is that a unique polynomial of a certain degree exists that accurately represents the function being approximated. In order to determine the coefficients of the interpolating polynomial, several methods can be employed, such as Lagrange interpolation, Newton’s divided difference interpolation, or Vandermonde matrices. These approaches aim to calculate the coefficients that yield a polynomial satisfying the condition of passing through all the given data points.

### 4.3. Metrics

We used the mean absolute error (MAE), mean relative error (MRE), and root-mean-square error (RMSE) to evaluate the performance of the algorithms. These metrics were calculated as follows: (27)$$\mathrm{MAE}=\frac{1}{n}\sum_{i=1}^{n}\left|y_{i}-x_{i}\right|.$$
(28)$$\mathrm{MRE}=\frac{1}{n}\sum_{i=1}^{n}\left|\frac{x_{i}-y_{i}}{y_{i}}\right|.$$
(29)$$\mathrm{RMSE}=\sqrt{\frac{1}{n}\sum_{i=1}^{n}{\left(y_{i}-x_{i}\right)}^{2}}.$$
where *n* represents the number of samples, $y_{i}$ represents the observed value, and $x_{i}$ represents the predicted value.

## 5. Results and Discussion

We conducted reconstruction experiments using multiple algorithms on two datasets. We focused on assessing the reconstruction accuracy and stability of the XGBoost-LSTM algorithm. Additionally, we examined the robustness of the algorithm and the accuracy loss with a reduction in the number of temperature sensors.

### 5.1. Reconstruction Results for Temperature Field of Sensor Head

The experimental computations were performed in an environment utilizing PyTorch 2.0.1-cu117 and Python 3.11 on a hardware framework enriched with an Intel Core i7 10870H CPU and an Nvidia RTX 3080 GPU. The operating system used was Windows 11, version 21H2. The hyperparameters of each model were adjusted and tested multiple times to achieve near-optimal reconstruction results.

Their values are presented in Table 1.

#### 5.1.1. Reconstruction Results of Ground Test Data

First, we compared the overall performance of the models, as shown in Figure 13. The reconstruction results of the three models presented certain fluctuations in different areas. The PI algorithm exhibits the maximum averaged error (696 μK) at D3 and the minimum averaged error (523 μK) at C4. The BP algorithm exhibits the maximum average error (63.7 μK) at D3 and the minimum average error (48.3 μK) at C2. XGBoost-LSTM exhibits the maximum average error (7.83 μK) at D4 and the minimum average error (7.4 μK) at E1. Overall, the XGBoost-LSTM algorithm achieved a reconstruction accuracy of two orders of magnitude higher than that of PI and one order of magnitude higher than that of the BP neural network algorithm.

In addition, we found that the reconstruction results of XGBoost-LSTM are not only high in accuracy but are also stable, showing consistent reconstruction accuracy at all positions. This may be due to the fact that the weights of the temperature sensors were obtained during the pre-training process, thus achieving accurate predictions in each area.

Table 2 presents the reconstruction results for the algorithms. For all the evaluation metrics, XGBoost-LSTM outperformed the other algorithms. This is due to the ability of LSTM to further learn the dynamic changes in time-series data and to capture the evolution of sensor readings over time through memory units, thereby achieving more accurate predictions in the reconstruction of temperature fields.

Next, we compared the detailed information of the residuals in each area, where the residuals are represented by absolute errors. In Figure 14, the overall performance of the PI algorithm is poor, and its residual distribution is unacceptable in both the best and worst cases. For example, in the D4 area, the maximum error is 2.8 mK. Similar results are observed in other areas.

In comparison, the BP neural network exhibits better performance with regard to the average residual in each area, as shown in Figure 15. However, in the residual plot, the BP neural network exhibits varying amounts of abnormal errors in different areas, with some errors reaching 0.25 mK. This implies that the model did not fully explain the data. When the data changes or when we need to extrapolate predictions for complex real-world data, it is likely to be ineffective.

Next, we examine the performance of the XGBoost-LSTM algorithm. In Figure 16, the reconstruction errors in each area are small, indicating that the algorithm accurately reconstructed each area. This is because the pre-training of XGBoost plays a role in adaptively adjusting the weights for predictions in different areas, yielding different parameter effects in different areas.

#### 5.1.2. Reconstruction Results for On-Orbit Test Data

We examined the overall reconstruction effectiveness for various algorithms. Because the PI interpolation algorithms were ineffective, we excluded them. Here, we focus on comparing the BP neural network and XGBoost-LSTM algorithms. In Figure 17, owing to the distances of the temperature sensors from the EH, both algorithms experience a certain degree of accuracy decline. However, the XGBoost-LSTM algorithm maintained a significant advantage over the BP neural network with regard to reconstruction accuracy in various areas, and its accuracy was consistent among the different areas. This, once again, demonstrates the superiority of XGBoost in feature selection. The results are presented in Table 3.

From the data in the table, as the data becomes more complex, the reconstruction accuracies of the algorithm all decline. However, the difference between the maximum error and the minimum error of the reconstruction of XGBoost-LSTM is still smaller. This is because the design of the XGBoost-LSTM algorithm incorporates the idea of ensemble learning, which can improve prediction accuracy by building multiple decision trees and considering their interaction.

We also examined the detailed reconstruction residuals of various areas using on-orbit data. The reconstruction performance of different algorithms is shown in Figure 18 and Figure 19.

The prediction performance of the BP neural network fluctuated significantly. In the best and worst-case scenarios, the prediction accuracy differed by 15.4 and 166 μK for the ground and on-orbit data, respectively. For comparison, the XL algorithm exhibited accuracy differences of 0.43 and 20.4 μK, respectively. The XL algorithm exhibited not only accuracy but also remarkable stability in predicting the temperatures in various areas. This may be because when the reliability of the input data decreases, XGBoost-LSTM adjusts the weights of input features and feature crosses. Considering that some temperature sensors are too close to a major heat source but far from the EH, the reading influence of temperature sensors on the EH may change.

Therefore, the BP neural network fails, while XGBoost-LSTM can determine which features are important. The intermediate process of the algorithm is displayed in Figure 20, where we observe that the sensitivities of each area to different temperature sensors are different. These data are used for feature crossing; thus, the sensor data allow the model to reconstruct the temperature in each area accurately. Figure 21 illustrates a similar situation with ground test simulation data.

### 5.2. Discussion of Reconstruction Accuracy

We further analyze the amplitude spectral density to verify that this new method meets the technical requirements of the Taiji project. As mentioned previously, the surface temperature gradient of the EH is due to the influence of heat sources inside the spacecraft. Our task primarily focuses on the effects of noise, with a frequency of ≥0.1 mHz on GRS. For the ground test simulation data, we found that the reconstruction effect of the XL algorithm is best in the E1 area and slightly worse in the D4 area, with the reconstruction accuracy of 10 areas exceeding the average. For the on-orbit test simulation data, the XL algorithm has the best reconstruction effect in the B3 area and is slightly less effective in the A3 area, with the reconstruction accuracy of 12 areas exceeding the average. Hence, we selected these areas for an error spectrum analysis to verify that our reconstruction model fulfills the temperature noise budget requirements of the Taiji project. As shown in Figure 22, we investigated the error amplitude spectral density (ASD) diagrams.

In order to further evaluate the effectiveness of the algorithm for reconstructing the temperature of the EH, we illustrate our assessment with an example of moderate reconstruction results for the on-orbit situation, as shown in Figure 23. According to the ASD curve, in the medium- and low-frequency bands, the reconstruction accuracy is approximately 30 μK/$\sqrt{\mathrm{Hz}}$. Together with Equation (Equation 7), we used the following parameters to calculate the acceleration noise caused by the thermal effect.
(30)$$\begin{array}{cc} T_{0} & =300\;K\\ A & =2.116\times 10^{-3}\;m^{2}\\ P & =1\times 10^{-5}\;\mathrm{Pa}\\ m_{t} & =2\;\mathrm{kg}\\ c & =3\times 10^{8}\;{\mathrm{ms}}^{-1}\\ \sigma & =5.670373\times 10^{-8}\;W\cdot m^{-2}\cdot K^{-4} \end{array}$$

In this situation, the acceleration noise is approximately $10^{-16}{\mathrm{ms}}^{-2}{\mathrm{Hz}}^{-1/2}$.

This means that we can use the temperature data of the EH surface to calibrate the aforementioned thermal noise and subtract it in the subsequent data processing.

When calculating thermal noise, the temperature change on each surface of the EH was averaged, and this may overlook important details. In contrast, we divided the surface of the EH into areas and reconstructed high-accuracy temperature data for up to 24 areas on the surface of the EH to obtain a more detailed temperature distribution. This provides more information and allows for more accurate measurements of the changes in radiation pressure; thus, the torque impact of each divided area on the TM can be determined, obtaining a more accurate estimate of the total torque. Based on the estimated acceleration noise, we assumed that the distance from the TM to the torsion axis, i.e., R, is 23 mm. This leads to a theoretical torque resolution derived from noise predictions at $10^{-17}{\mathrm{NmHz}}^{-1/2}$. It is important to note that this resolution represents an ideal limit based on noise characteristics rather than the actual measurement accuracy of the torsion balance system itself, which may be subject to additional constraints.

### 5.3. Discussion of Robustness of Algorithm

As described earlier, the number of temperature sensors is limited; therefore, we must reconstruct the temperature of the sensor head using a limited number of temperature sensors. Therefore, the number of sensors is an important factor, and we investigated the accuracy loss of the algorithm when reducing the number of sensors. In Figure 24, with the reduction in the number of sensors, the loss in reconstruction accuracy of the XL algorithm is minimized. When the number of sensors decreased by 50%, the accuracy only decreased by 2%. In contrast, the accuracy loss of the BP neural network reached 400%. Thus, the proposed algorithm can be used to determine the optimal number of sensors, and its reconstruction capability is only slightly weakened when a small number of temperature sensors fail.

It should be noted that the location of sensors can also influence the experimental results. The analysis presented in this study is based on a sequential reduction in the number of temperature sensors (i.e., G_T1, G_T2, G_T3, etc.) to explore the impact of sensor quantity reduction on the algorithm’s reconstruction performance. In subsequent research, we plan to investigate the sensitivity of the reconstruction algorithm to the positioning of sensors further.

## 6. Conclusions

In the Taiji project, high-precision GRSs are the key payload for low-frequency space-borne gravitational wave detection. The sensor heads are subjected to stray forces caused by temperature disturbances. By considering spatial constraints and sensor coupling relationships, we developed an innovative algorithm called XGBoost-LSTM to reconstruct the temperature on the surface of the EH using a limited number of temperature sensors. In the ground test data reconstruction, the algorithm’s reconstruction error is at the level of $10^{-6}$ K, whereas for on-orbit test data reconstruction, the algorithm’s reconstruction error is at the level of $10^{-5}$ K. The amplitude spectral density plot of the reconstruction error indicated that its error was of the order of $10^{-5}$
${\mathrm{KHz}}^{-1/2}$. This reconstruction accuracy satisfies the key payload temperature control target of the Taiji project, which is 100 μ${\mathrm{KHz}}^{-1/2}$.

The reconstruction algorithm we used has the following advantages: First, we can reconstruct the 24 areas of the EH surface with high accuracy. Through the calculation of the temperature of multiple areas, we can model the thermal effects mentioned in the previous text more accurately, estimate the stray force produced by each area, and, thus, obtain the total thermal noise or total torque. Second, under two types of simulated data, the reconstruction accuracy of the XL algorithm is higher than BP and meets the task requirements. When facing the complex environment of space, our reconstruction algorithm will be more reliable than other methods. Third, XL has high robustness, which means that the algorithm will not overly rely on more temperature sensors. In addition, if a small number of temperature sensors fail, the loss of the reconstruction result will not be too large. Finally, the method we propose is not limited to the temperature reconstruction of the sensor head of GRSs and can be used as a solution for the temperature field reconstruction problem for other complex structures and precise measurement backgrounds.

In the next phase, we will optimize the algorithm using experimental data and, in conjunction with experimentation, explore the optimal number and placement of temperature sensors. Additionally, we will conduct a more detailed division of an EH to enhance the reconstruction accuracy and stability of the algorithm in various areas.

## Figures and Tables

**Figure 1 sensors-24-02529-f001:**
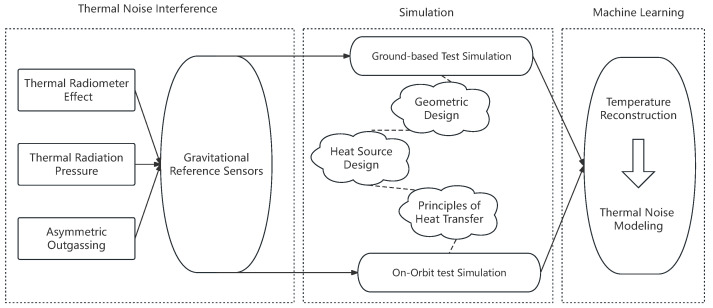
This paper first introduces three types of thermal effects, followed by simulations of the sensor head using different design schemes. Subsequently, a machine learning model is trained to reconstruct the temperature distribution of the sensor head, thereby enabling the modeling and estimation of thermal noise.

**Figure 2 sensors-24-02529-f002:**
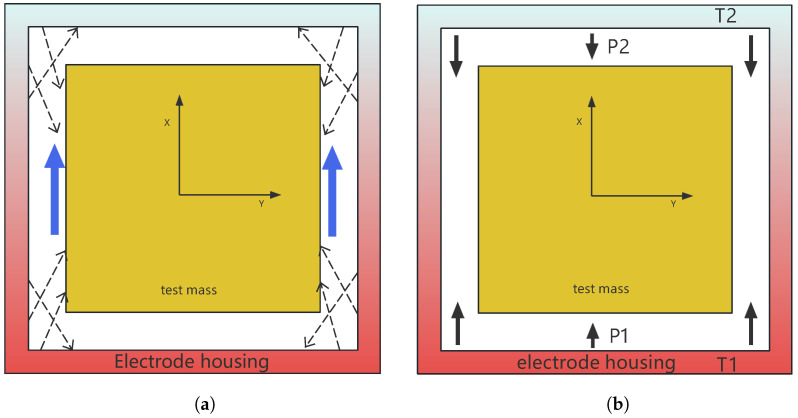
(**a**) This image depicts a cross-section of a GRS. When there is a temperature gradient along the *x*-axis, the red color represents the hot area, and the blue color represents the cold area. The presence of this thermal gradient causes a net thermal shear force (blue arrows) along the *x*-axis. The dashed arrow represents the emission of molecules from a specific point on the surface of the EH. (**b**) Due to the different temperatures on the upper and lower surfaces of the EH, different magnitudes of pressure are exerted on the respective faces, affecting the TM.

**Figure 3 sensors-24-02529-f003:**
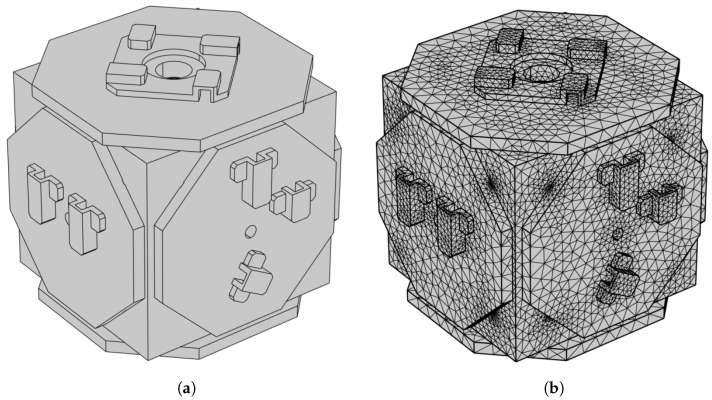
(**a**) The geometric design of the EH, which is made up of aluminum and sapphire materials; (**b**) the mesh division of finite element simulation.

**Figure 4 sensors-24-02529-f004:**
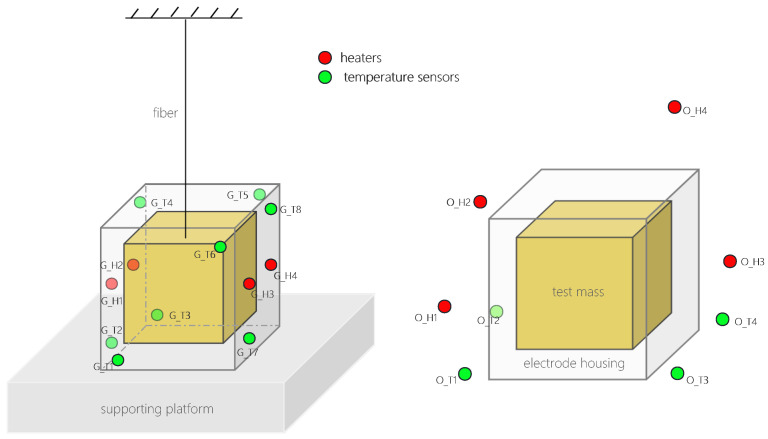
The red dots represent the locations of the heating heat source, and the green dots represent the locations of the temperature sensors. On the left side, in the simulated ground-based experiment, the EH is alternately heated by symmetric heat sources on both sides, and the temperature sensor is located on the surface of the EH. On the right side, in the simulated on-orbit test experiment, some random heat sources transfer heat to the EH through thermal radiation, and the temperature sensor is located around the EH.

**Figure 5 sensors-24-02529-f005:**
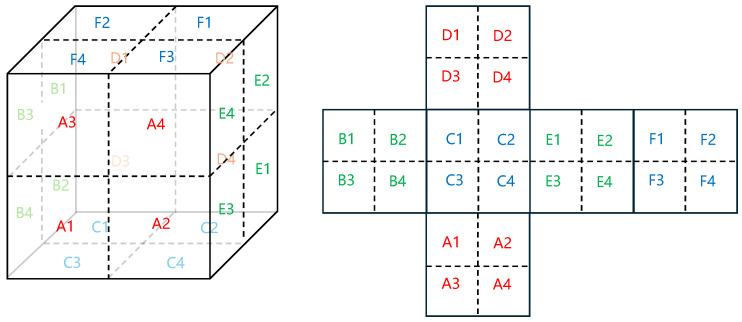
This is a simplified EH and its unfolded diagram, with each surface divided into four zones that are numbered. We assume that the temperature in each area is the same, using the temperature at the central position of each zone to represent the temperature gradient potential of each surface.

**Figure 6 sensors-24-02529-f006:**
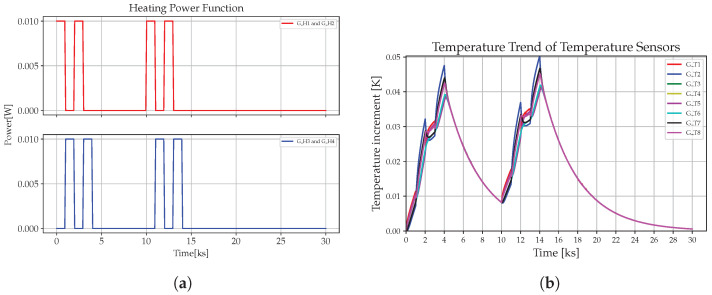
(**a**) The heating function of the heat source; (**b**) the change in temperature of various temperature sensors.

**Figure 7 sensors-24-02529-f007:**
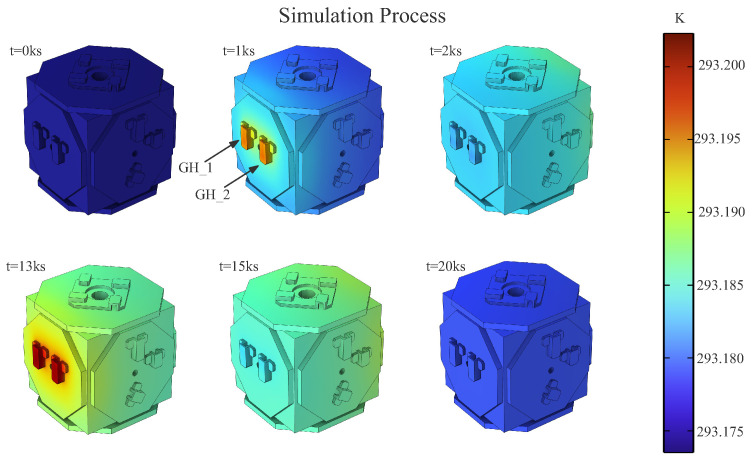
With the alternate heating of the heaters, the surface temperature of EH changes. Another set of heaters—GH_3, GH_4—are positioned on the other side of the EH.

**Figure 8 sensors-24-02529-f008:**
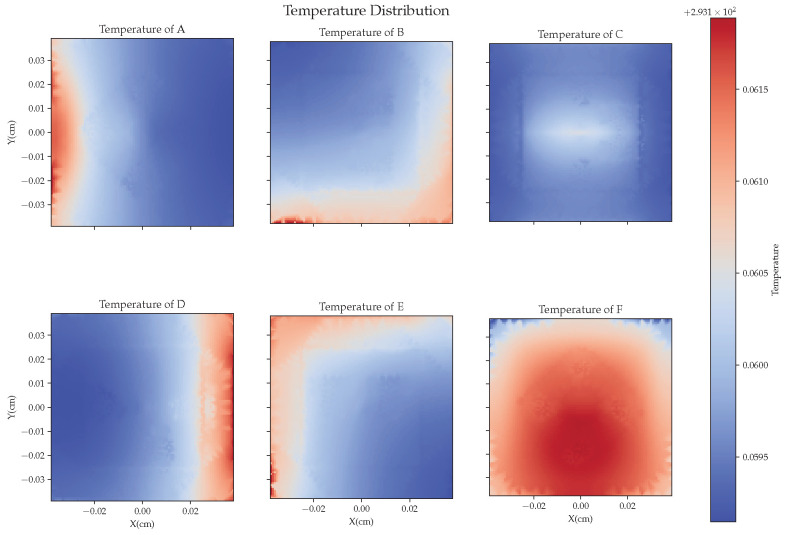
The surface temperature distribution of the EH at a certain moment.

**Figure 9 sensors-24-02529-f009:**
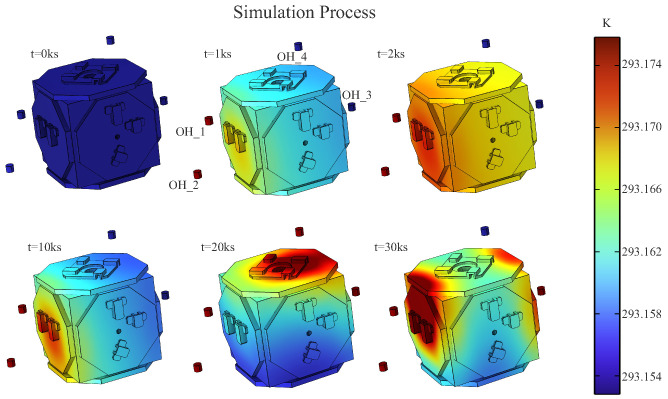
Four heat sources radiate towards the EH in a random manner, the image shows the temperature-changing process of the EH.

**Figure 10 sensors-24-02529-f010:**
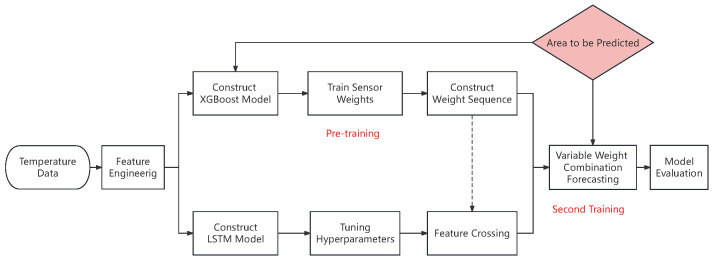
Firstly, the weight data of a temperature sensor were used for pre-training; then, these were cross-referenced with the original data through element-wise multiplication and finally input into LSTM (long short-term memory) for secondary training.

**Figure 11 sensors-24-02529-f011:**
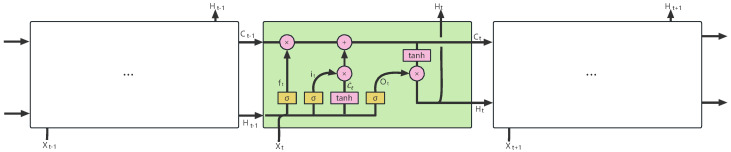
Diagram of LSTM structure.

**Figure 12 sensors-24-02529-f012:**
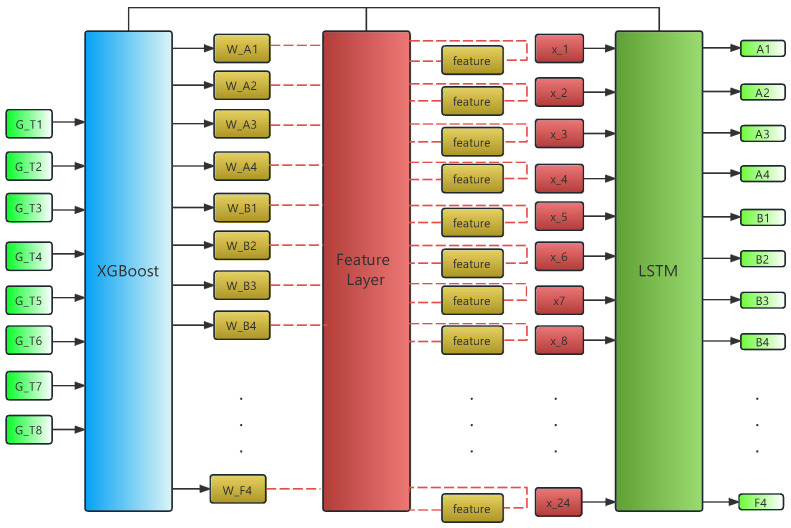
The figure illustrates the end-to-end learning process of XGBoost-LSTM. After the temperature sensor data are input into the model, it is first pre-trained by XGBoost. The training result contains the weight information of the temperature sensors. This weight information is then processed and crossed with the original data, which are subsequently input into LSTM for secondary learning. This process enables adaptive weight adjustment learning strategies for different areas.

**Figure 13 sensors-24-02529-f013:**
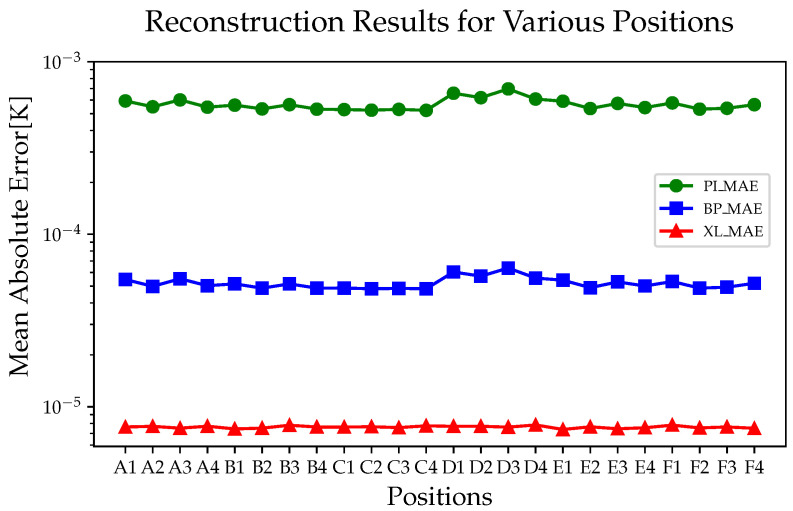
The average performance of different algorithms in different areas.

**Figure 14 sensors-24-02529-f014:**
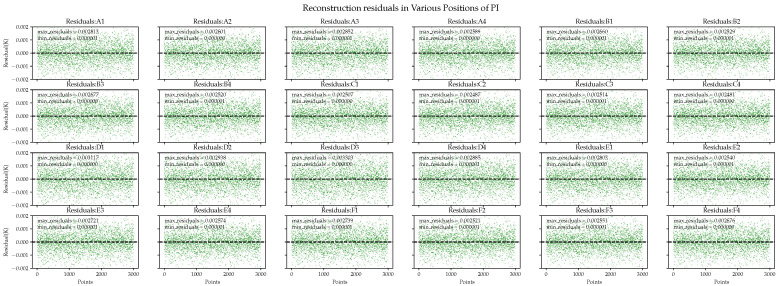
Polynomial interpolation performance in the reconstruction residuals of ground simulation data.

**Figure 15 sensors-24-02529-f015:**
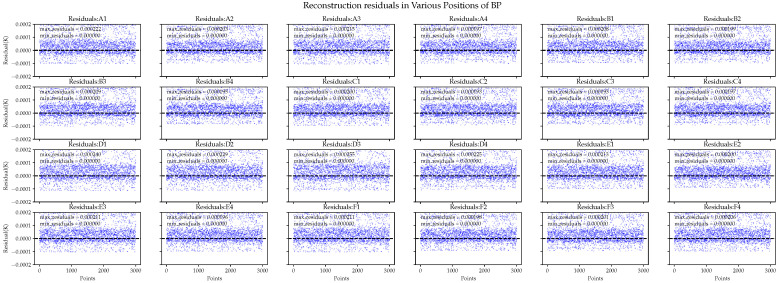
BP neural network performance in the reconstruction residuals of ground simulation data.

**Figure 16 sensors-24-02529-f016:**
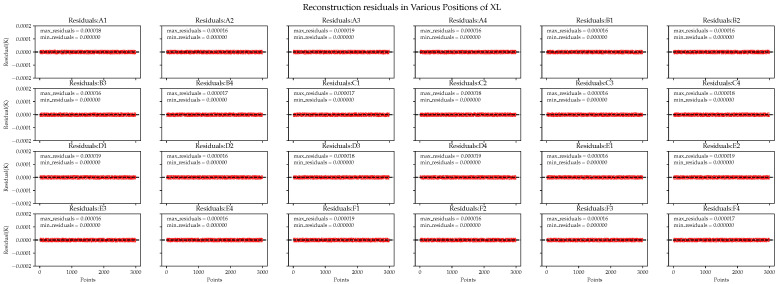
XGBoost-LSTM performance in the reconstruction residuals of ground simulation data.

**Figure 17 sensors-24-02529-f017:**
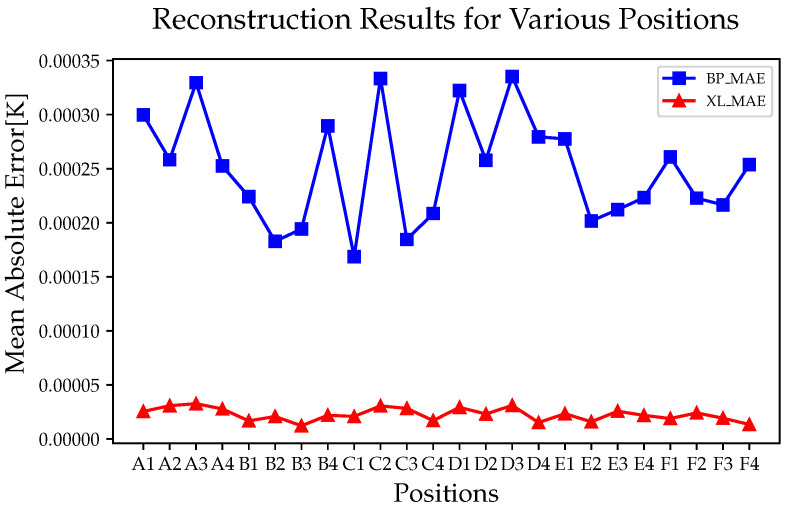
The average performance of different algorithms at different areas.

**Figure 18 sensors-24-02529-f018:**
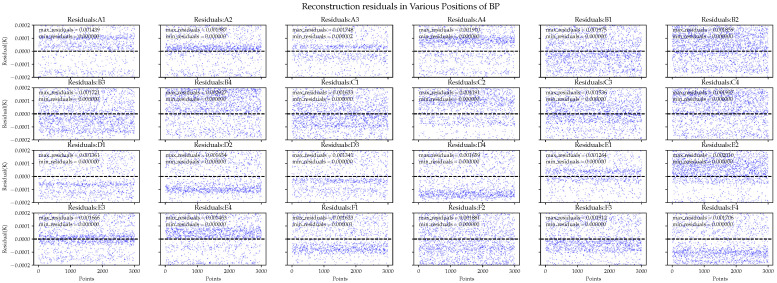
BP neural network performance in the reconstruction residuals of the on-orbit simulation data.

**Figure 19 sensors-24-02529-f019:**
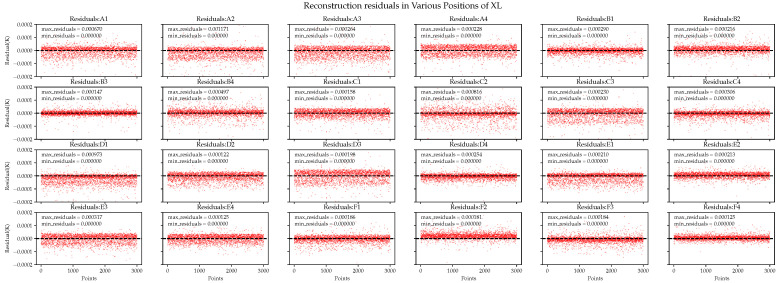
XGBoost-LSTM performance in the reconstruction residuals of the on-orbit simulation data.

**Figure 20 sensors-24-02529-f020:**
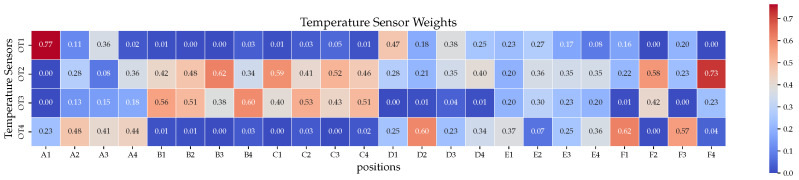
The importance of temperature sensors in different areas (on-orbit data).

**Figure 21 sensors-24-02529-f021:**
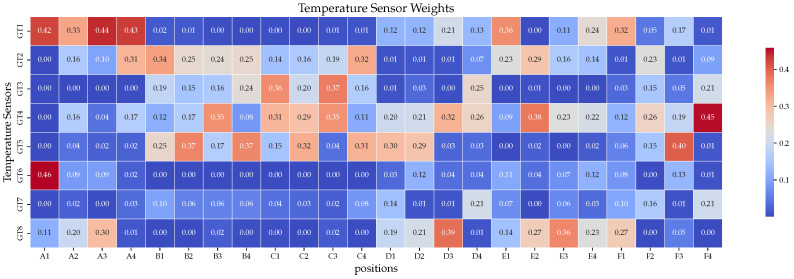
The importance of temperature sensors in different areas (ground data).

**Figure 22 sensors-24-02529-f022:**
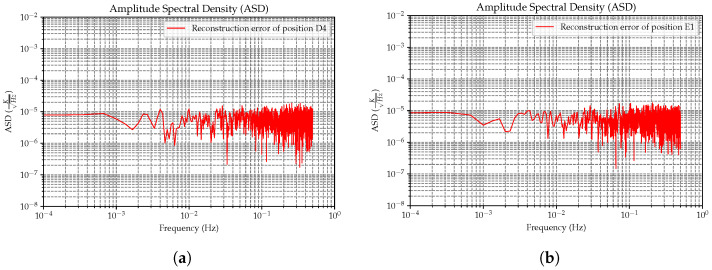
(**a**,**b**) The amplitude spectrum density images of the residuals in the less-optimal and best-case scenarios, respectively, of the reconstruction algorithm in the ground test data. (**c**,**d**) The amplitude spectrum density images of the residuals in the less-optimal and best-case scenarios of the reconstruction algorithm, respectively, in the on-orbit test data.

**Figure 23 sensors-24-02529-f023:**
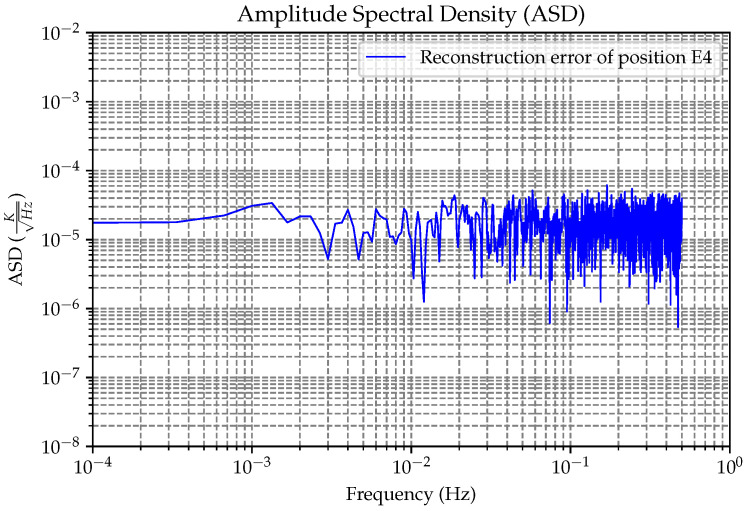
For the amplitude spectral density image of the general results of the on-orbit reconstruction, we used this result to estimate the recognition level of thermal noise.

**Figure 24 sensors-24-02529-f024:**
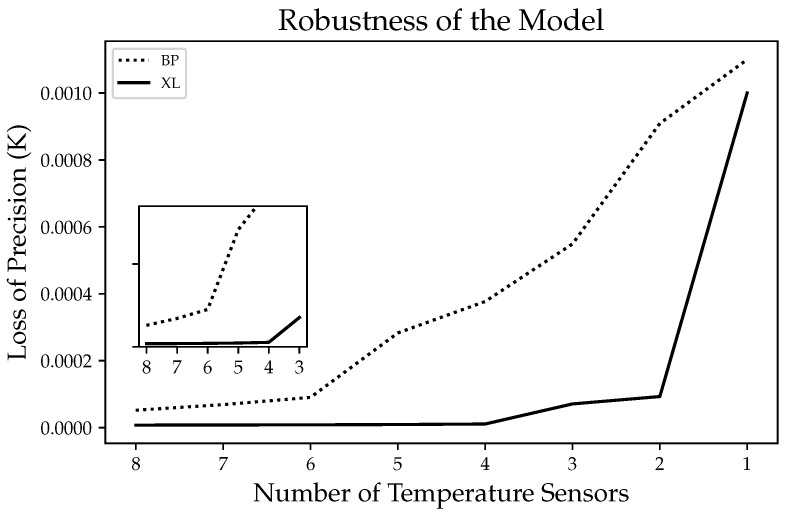
The loss of reconstruction accuracy (MAE) of the BP neural network and XGBoost-LSTM when the number of temperature sensors decreases.

**Table 1 sensors-24-02529-t001:** The parameters involved in the experiment, including those of XGBoost-LSTM and the baseline models.

Parameters	Values
Data-partitioning method ^1^	8:1:1
Optimizer ^2^	adam
Learning rate ^2^	0.01
Activation ^2^	linear
XL-max_depth	12
XL-n_estimators	500
LSTM-No. layers	5
LSTM-droupout	0.2
LSTM-trainable params	34,024
BP-No. layers	3
BP-trainable params	3704

^1^ The values in “8:1:1” represent the ratios of the training set, validation set, and test set, respectively. ^2^ The parameter values indicated are utilized in the corresponding models. The parameters shared across multiple models use the same values in each model.

**Table 2 sensors-24-02529-t002:** Reconstruction results for the models in various areas.

Area	MAE (μK)			RMSE (μK)			MRE ($\times 10^{-6}\%$)		
	PI	BP	XL	PI	BP	XL	PI	BP	XL
A1	593	54.6	7.63	763	76.0	8.79	202	18.6	2.6
A2	548	49.9	7.69	705	69.7	8.86	187	17.0	2.62
A3	601	55.2	7.5	773	77.0	8.7	205	18.8	2.56
A4	546	50.3	7.71	702	70.1	8.84	186	17.2	2.63
B1	560	51.5	7.43	721	71.6	8.65	191	17.6	2.53
B2	533	48.7	7.49	686	68.0	8.67	182	16.6	2.56
B3	564	51.5	7.81	726	71.9	8.99	192	17.6	2.66
B4	531	48.7	7.64	683	67.9	8.84	181	16.6	2.61
C1	528	48.7	7.62	679	68.1	8.83	180	16.6	2.6
C2	524	48.3	7.65	674	67.3	8.8	179	16.5	2.61
C3	530	48.5	7.58	681	67.7	8.8	181	16.5	2.59
C4	523	48.3	7.74	672	67.6	8.88	178	16.5	2.64
D1	657	60.4	7.71	845	84.4	8.91	224	20.6	2.63
D2	619	57.2	7.69	796	80.1	8.89	211	19.5	2.62
D3	696	63.7	7.61	895	89.2	8.77	237	21.7	2.6
D4	608	55.7	7.83	782	77.6	9.02	207	19.0	2.67
E1	591	54.1	7.4	760	75.8	8.64	201	18.5	2.52
E2	536	49.0	7.64	689	68.3	8.8	183	16.7	2.6
E3	574	52.9	7.44	738	74.0	8.61	196	18.1	2.54
E4	542	50.1	7.56	698	69.9	8.75	185	17.1	2.58
F1	577	53.2	7.81	743	74.0	8.97	197	18.1	2.67
F2	531	48.7	7.54	683	67.9	8.72	181	16.6	2.57
F3	537	49.4	7.64	691	69.2	8.84	183	16.8	2.6
F4	564	51.9	7.49	725	72.2	8.67	192	17.7	2.56
AVG	567	52.1	7.61	729	72.7	8.80	193	17.7	2.59
MIN	523	48.3	7.4	672	67.3	8.61	178	16.5	2.52
MAX	696	63.7	7.83	895	89.2	9.02	237	21.7	2.67

**Table 3 sensors-24-02529-t003:** Reconstruction results of the models in various areas.

Area	MAE (μK)		RMSE (μK)		MRE ($\times 10^{-6}\%$)	
	BP	XL	BP	XL	BP	XL
A1	300	25.4	392	38.9	102	8.68
A2	258	30.8	365	54.1	88.1	10.5
A3	330	32.6	446	47.6	112	11.1
A4	252	27.8	372	34.7	86.1	9.47
B1	224	16.7	391	25.9	76.4	5.71
B2	183	20.8	320	29.8	62.3	7.09
B3	194	12.2	301	18.1	66.2	4.16
B4	290	22.0	521	37.0	98.8	7.49
C1	169	20.9	266	27.7	57.5	7.12
C2	333	30.6	633	59.4	114	10.4
C3	185	28.3	290	39.1	62.9	9.64
C4	209	17.0	343	26.2	71.2	5.79
D1	322	29.2	420	48.3	110	9.97
D2	258	23.1	349	31.1	87.9	7.87
D3	335	31.1	441	39.3	114	10.6
D4	279	15.2	376	21.4	95.3	5.17
E1	278	23.4	371	33.4	94.7	7.98
E2	202	15.9	357	20.8	68.8	5.41
E3	212	25.7	323	33.1	72.3	8.75
E4	223	21.9	325	27.5	76.1	7.46
F1	261	18.9	356	27.5	89.0	6.45
F2	223	24.1	362	30.5	76.0	8.22
F3	217	19.2	328	28.4	73.9	6.56
F4	254	13.4	357	18.3	86.5	4.57
AVG	249	22.7	375	33.2	85.0	7.75
MIN	169	12.2	266	18.1	57.5	4.16
MAX	335	32.6	633	59.4	114	11.1

## Data Availability

The data underlying the results presented in this paper are not publicly available at this time but may be obtained from the authors upon reasonable request.

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
