# Peer review of "Sensor Head Temperature Distribution Reconstruction of High-Precision Gravitational Reference Sensors with Machine Learning"

_sensors, 2024, doi:10.3390/s24082529_

Round 1

Reviewer 1 Report

Comments and Suggestions for Authors

The authors try to employ the XGBoost-LSTM model to reconstruct the temperature distribution over the sensor head of electrostatic gravitational reference sensors (GRS), which could be more accurate than the traditional interpolation method. This new method is quite interesting and could effectively improve the estimation accuracy of the key noises for both the cases of ground-based performance tests and in-orbit experiments. While, before considering for publication in the journal of Sensors, I suggest the authors to answer the following comments and make major revisions.

1.       Considering the readability of this manuscript, I found some parts are rather wordy and tedious, and even confusing, e.g., the contents in lines 82 to 88, lines 112 to 122, lines 171 to 172 and so on. I suggest the authors to carefully revise the manuscript to improve the readability.

2.       In line 116 of section 3.1, “a simplified EH model” is defined, how does this model differ from the real EH of the GRS considered? What simplification has been made?

3.       In Figure 3, is the deployment of temperature sensors based on the real settings of ground tests or in-orbit experiments?

4.       Detailed information of the total number of the temperature sensors, and also the relative locations of each temperature sensor and heater to the sensor head should be clarified.

5.       In page 7, line 225, why the temperature variation range in the simulation for the case of in-obit experiments is set from 293.150K to 293.216 K? Such temperature variation seems rather large compare to the requirements of temperature control for space-borne gravitational wave antennas.  This needs some more explanations.

6.       In figure 11 the unit is missing.

7.       In table 1, what do you mean by "Data partitioning method"?

Reviewer 2 Report

Comments and Suggestions for Authors

Dear Authors,

Firstly, I would like to express my gratitude to you for such excellent presentation of the results for this research. Compensation of temperature noise for gravity sensors or other sensors of physical quantities is extremely important. You have excellent motivation for research and a solid evidence base, which is rare in publications. Thank you. But I still have very few questions about your text, which I hope will help your future Readers understand your research in more detail. There are few questions and I left them in the attached file "sensors-2915509-peer-review-v1 (Review)". I will be waiting for your answer.

Kind reagrds,

Reviewer

Reviewer 3 Report

Comments and Suggestions for Authors

The manuscript introduces the XGBoot-LSTM algorithm for sensor head temperature reconstruction, a crucial aspect of the Taiji gravitational wave detection program. The algorithm demonstrates high accuracy and robustness, making it a valuable contribution to the field. Overall, the manuscript is well-written, and the conclusions are supported by solid results. I recommend acceptance of the manuscript after addressing a few minor issues.

  1. 1. The manuscript would benefit from additional illustrations explaining the background and problem statements, particularly to make it more accessible to a general audience.

  2. 2. In Figure 22, the model shows robust performance with a small number of failed temperature sensors. However, it would be valuable to discuss how the location of the failed sensors might impact the prediction accuracy.

  3. 3. There are some typos present in the manuscript. For instance, in section 2, the three temperature effects should be labeled as subtitles 2.1, 2.2, and 2.3 respectively, as "2.1 Thermal radiometer effect", "2.2 Thermal radiation pressure", and "2.3 Asymmetric outgassing".

Comments on the Quality of English Language

The quality of English language in the manuscript is generally good. The sentences are well-structured, and the ideas are effectively communicated. However, there are still some typos.

Reviewer 4 Report

Comments and Suggestions for Authors

The paper’s scope is within the scope of the journal, and it presents an original contribution. The abstract is sufficient to give useful information about the paper’s topic. The proposed algorithm is described. The paper is well-structured and written, and the text is clear and easy to read. However, there are some comments we recommend the authors to respond to:

Comment-1: In the introduction section or where appropriate, you may need to cite and add the following recent reference regarding High-Precision Inertial Sensors:

Liu, Y.; Yu, T.; Wang, Y.; Zhao, Z.; Wang, Z. High-Precision Inertial Sensor Charge Ground Measurement Method Based on Phase-Sensitive Demodulation. Sensors 2024, 24, 1009. https://doi.org/10.3390/s24031009

Comment-2: In Section 3 and before Subsection 3.1, write one small overview paragraph about Section 3 and its subsections.

Comment-3: In Section 3.2 and before Subsection 3.2.1, write one small overview paragraph about Section 3.2 and its subsections.

Comment-4: In Section 4 and before Subsection 4.1, write one small overview paragraph about Section 4 and its subsections.

Comment-5: It is more appropriate to present the proposed algorithm as either pseudocode or flowchart and accordingly briefly explain it.

Comment-6: Since the proposed algorithm uses LSTM and machine learning, does the proposed algorithm suffer from a vanishing gradient problem? Could you please elaborate more regarding this issue and accordingly cite the following reference:

I. Abuqaddom, B. A. Mahafzah, H. Faris “Oriented stochastic loss descent algorithm to train very deep multi-layer neural networks without vanishing gradients” Knowledge-Based Systems, Vol. 230, Article 107391, 2021. https://doi.org/10.1016/j.knosys.2021.107391

Comment-7: In Section 3 or where appropriate, you need to present the experimental environment from hardware specifications (machine specifications), operating system, and programming language/tool.

Comment-8: The obtained results in Figures 11 and 15 and Tables 2 and 3 must be explained and justified according to the algorithms’ design point of view.

Comment-9: At the end of the conclusion section, it is worthwhile to elaborate more about the best-obtained results as values/percentages.

Comments on the Quality of English Language

English Language: The quality of the English language is good. The authors may need to check the whole manuscript for grammar, spelling, and formatting issues in general.

Round 2

Reviewer 2 Report

Comments and Suggestions for Authors

Dear Authors,

Thank you for your detailed response. I wish you good luck in your future research.

Kind regards,

Reviewer